# Muscle Wasting and Treatment of Dyslipidemia in COPD: Implications for Patient Management

**DOI:** 10.3390/biomedicines13081817

**Published:** 2025-07-24

**Authors:** Andrea Bianco, Raffaella Pagliaro, Angela Schiattarella, Domenica Francesca Mariniello, Vito D’Agnano, Roberta Cianci, Ersilia Nigro, Aurora Daniele, Filippo Scialò, Fabio Perrotta

**Affiliations:** 1Department of Translational Medical Sciences, University of Campania Luigi Vanvitelli, 80131 Naples, Italy; angela.schiattarella1@studenti.unicampania.it (A.S.); nikamariniello93@gmail.com (D.F.M.); vito.dagnano@studenti.unicampania.it (V.D.); cianciroberta@libero.it (R.C.); fabio.perrotta@unicampania.it (F.P.); 2Unit of Respiratory Medicine “Luigi Vanvitelli”, A.O. dei Colli, Monaldi Hospital, 80131 Naples, Italy; 3Department of Pharmaceutical, Biological, Environmental Sciences and Technologies, University of Campania “Luigi Vanvitelli”, Via G. Vivaldi 42, 81100 Caserta, Italy; ersilia.nigro@unicampania.it; 4CEINGE-Biotechnologies Advances Scarl, Via G. Salvatore 486, 80145 Naples, Italy; aurora.daniele@unina.it (A.D.); filippo.scialo@unina.it (F.S.); 5Department of Molecular Medicine and Medical Biotechnologies, University of Naples Federico II, 80131 Naples, Italy

**Keywords:** Chronic Obstructive Pulmonary Disease (COPD), comorbidities, sarcopenia, muscle wasting, dyslipidemia

## Abstract

Chronic Obstructive Pulmonary Disease (COPD) is a multifactorial condition associated with significant systemic complications such as cardiovascular disease (CVD), metabolic disorders, muscle wasting, and sarcopenia. While Body Mass Index (BMI) is a well-established indicator of obesity and has prognostic value in COPD, its role in predicting disease outcomes is complex. Muscle wasting is prevalent in COPD patients and exacerbates disease severity, contributing to poor physical performance, reduced quality of life, and increased mortality. Additionally, COPD is linked to metabolic disorders, such as dyslipidemia and diabetes, which contribute to systemic inflammation and worse prognosis and, therefore, should be treated. The systemic inflammatory response plays a central role in the development of sarcopenia. In this review, we highlight the mixed efficacy of statins in managing dyslipidemia in COPD, considering side effects, including muscle toxicity in such a frail population. Alternative lipid-lowering therapies and nutraceuticals, in addition to standard treatment, have the potential to target hypercholesterolemia, which is a coexisting condition present in more than 50% of all COPD patients, without worsening muscle wasting. The interference between adipose tissue and lung, and particularly the potential protective role of adiponectin, an adipocytokine with anti-inflammatory properties, is also reviewed. Respiratory, metabolic and muscular health in COPD is comprehensively assessed. Identifying and managing dyslipidemia and paying attention to other relevant COPD comorbidities, such as sarcopenia and muscle wasting, is important to improve the quality of life and to reduce the clinical burden of COPD patients. Future research should focus on understanding the relationships between these intimate mechanisms to facilitate specific treatment for systemic involvement of COPD.

## 1. Introduction

Chronic Obstructive Pulmonary Disease (COPD) is a heterogeneous lung condition characterized by persistent and progressive airflow obstruction [1,2]. COPD is associated with significant comorbidities, such as cardiovascular disease (CVD), metabolic disorders, cancer, osteoporosis, skeletal muscle dysfunction, anxiety/depression, cognitive impairment, gastrointestinal (GI) diseases, and muscle wasting [3,4,5]. The systemic effects of COPD not only compromise pulmonary function but also exacerbate the risk of CVD, leading to increased morbidity and mortality [6]. Indeed, in patients with COPD who also have multiple comorbidities, the risk of mortality is greater due to the comorbidities rather than the severity of the COPD itself [7]. Cardiovascular disease represents a significant concern for patients with COPD, playing a substantial role in the elevated mortality rates associated with this condition [8]. Recent research highlights dyslipidemia as a significant comorbidity among COPD patients, suggesting its potential role in disease progression is through mechanisms, such as endothelial injury [9]. In this context, adiponectin, an adipocytokine with anti-inflammatory properties, may play a protective role in the cardiovascular health of COPD patients [10]. Although the impact of dyslipidemia on the progression of COPD is not yet fully understood, it represents a risk factor for the onset of cardiovascular disease in these individuals [9]. A comprehensive nationwide cohort study conducted in Taiwan has revealed a significant association between hyperlipidemia and an increased incidence of COPD among individuals [11]. Despite solid evidence for the efficacy of statins in reducing risk factors for CVD, its role in managing dyslipidemia and outcome improvement in COPD remains controversial and requires further investigation [12]. Understanding the relationship between cholesterol lowering strategies, muscle condition, and COPD may lead to improving clinical outcomes and enhance patient quality of life [13]. Dyslipidemia, muscle wasting, and sarcopenia are all expressions of systemic consequences of COPD and treatments are required to balance protection and side effects [14].

This review aims to explore the interplay between dyslipidemia, muscle wasting, and sarcopenia in COPD. Furthermore, as dyslipidemia has to be considered a modifiable risk factor that also requires treatment in COPD patients, we explored the effect of therapies in patients with muscle wasting and sarcopenia. In this context, we have considered the potential efficacy of alternative treatments, such as nutraceuticals and non-statin lipid-lowering agents, in preserving muscle mass whilst reducing cardiovascular risk factors and improving disease outcomes. A schematic study summary is represented in Figure 1.

### Methods

In this context, the objective of this narrative review is threefold: (i.) to investigate the interference of dyslipidaemia in muscle wasting and sarcopenia in COPD; (ii.) to assess the impact of therapies on patients with muscle wasting and sarcopenia; (iii.) to examine the potential of alternative treatments to statins, including nutraceuticals and non-statin lipid-lowering agents, in preserving muscle mass and improving clinical outcomes in this population. A comprehensive literature search was conducted by MEDLINE, Embase, and the Cochrane Database, including articles published in the last 20 years (from 2005 to 2025). Only the articles written in English were included. The following MESH terms were used, tailored to each of the three research aims: COPD, chronic obstructive pulmonary disease, dyslipidaemia, sarcopenia, muscle wasting, Hydroxymethylglutaryl-CoA Reductase Inhibitors, statin, Nutraceuticals, Nutraceutical, Dietary Supplement, Functional Food, Natural Compound, Polyphenol, and Omega-3 Fatty Acid. This review provides a narrative synthesis of the evidence identified in the literature search.

## 2. Cardiopulmonary Risk Associated with COPD: Mechanisms Contributing and Underlying Pathogenesis

COPD and CVD frequently coexist [12]. Among patients with COPD, the prevalence of cardiac disease ranges between 30% and 70% and a high proportion of patients with mild or moderate COPD die of cardiovascular events, including myocardial infarction, stroke, heart failure, or arrhythmia [13,14,15]. In patients with severe COPD, the deaths are mainly due to respiratory causes. Other than the increased risk of mortality, patients with COPD and CVD also have increased morbidity, worse quality of life and a greater number of hospitalizations with a longer length of stay [16,17]. COPD and CVD share common risk factors, such as smoking history, exposure to air pollution, more advanced age, physical inactivity, poor diet, or low socioeconomic status [18,19]. Furthermore, complex pathophysiological links exist between these two conditions.

Hyperinflation, characterized by elevated end-expiratory lung volume, is a hallmark of COPD, contributing to worse exertional dyspnea and exercise capacity. Airflow limitation may lead to lung hyperinflation with consequent increased falls in intrathoracic pressures, right-ventricular dysfunction, compromised left-ventricular filling, and reduced cardiac output [20,21]. In patients with COPD, chronic hypoxemia, due to a ventilation/perfusion mismatch, provokes pulmonary vasoconstriction and vascular remodeling, with augmented pulmonary vascular resistance and right-ventricular diastolic dysfunction [22]. Hypoxia and lung hyperinflation can cause pulmonary hypertension [23]. Hypoxia also increases systemic inflammation and oxidative stress [24]. Persistent low-grade systemic inflammation in COPD contributes to the formation and progression of atherosclerotic plaque. Several studies have shown that patients with stable COPD and CVD have higher levels of systemic inflammatory biomarkers, such as fibrinogen, C-reactive protein (CRP), interleukin (IL)-6, and IL-8 [25,26]. Levels of inflammatory biomarkers further increase during COPD exacerbation (AECOPD). A recent meta-analysis identified an increased risk of acute CV events after AECOPD, mainly severe, and in particular, the risk of acute myocardial was increased immediately after COPD exacerbation and remained high for up to a year later [27]. Possible mechanisms related to the risk of acute CV events after COPD exacerbation include systemic inflammation, acute hypoxemia, oxidative stress, and increased platelet activity [28,29]. These conditions may increase the risk of plaque rupture with subsequent acute cardiovascular event [30]. Another interesting finding is that patients with COPD have increased arterial stiffness, a strong predictive value for CV events, in comparison with age and smoking matched controls [31]. A possible explanation is that degradation of elastin occurs not only in the lung, resulting in emphysema, but also in the arterial walls, resulting in more pronounced arterial stiffness.

The presence of comorbid CVD inevitably complicates the management of COPD and specific recommendations are necessary for the management of CVD in patients with COPD. For example, β-blockers are widely prescribed in the treatment of CVD, but they are often underused in COPD patients due to their potential antagonism with β2-agonists inducing bronchoconstriction [32]. Cardio selective β-blockers (e.g., atenolol, bisoprolol, and metoprolol) should be recommended in patients with COPD. On the other hand, long-acting β2-agonists (LABAs) represent the mainstay pharmacological treatment of COPD, but they have been associated with an increased CV risk, possibly due to the stimulation of the sympathetic nervous system [33]. There are also data that suggest favorable effects of LABAs on CV risk, due to reducing lung hyperinflation and the rate of COPD exacerbations [34,35]. Similarly, long-acting muscarinic antagonists (LAMAs) reduce lung hyperinflation but they could induce cardiac arrhythmias, suppressing parasympathetic control of heart rate [36,37]. Other drugs routinely prescribed in patients with CVD are angiotensin conversion enzyme inhibitors (ACE-Is) and angiotensin receptor blockers (ARBs). A study suggests that the use of ACE-Is or ARBs slow the progression of emphysema, but a side effect of ACE-I is coughing [38]. In addition, an observational cohort study among patients with COPD found that ARBs were associated with lower risk of severe exacerbations and mortality, in comparison to ACEIs [39]. This finding may suggest that ARBs are a preferred choice for patients with COPD. Patients with COPD often receive statins for the prevention of cardiovascular risk. Many studies suggest positive effects of statin use in patients with COPD, reducing lung function decline and exacerbations and improving symptoms [40,41,42]. Antiplatelet therapy may be beneficial after an exacerbation when platelet activation occurs [43].

## 3. Additional Mechanisms Implicated in COPD: From Muscle Wasting to Sarcopenia

Body Mass Index (BMI) is an accepted indicator of obesity, and it is associated with COPD [15]. Patients with lower BMI had more severe COPD symptoms, including worse airflow obstruction, hyperinflation, and osteoporosis. Conversely, patients with an increase in BMI had a higher prevalence of cardiovascular comorbidities and systemic inflammation, indicating that adipose tissue might modulate disease expression differently in these two groups [16]. In a large cohort of the Korean population, the authors demonstrated that a decrease in BMI is associated with an increased risk of severe exacerbations and all-cause mortality in COPD patients. Notably, the relationship between BMI reduction and mortality is dose dependent. Conversely, an increase in BMI is linked to a higher risk of death only among obese COPD patients [17]. Abnormal lipid metabolism in COPD patients can impair immunity, airway repair, and tissue remodeling, while excessive fat accumulation worsens inflammation and metabolic dysfunction [18]. These findings highlight the importance of monitoring BMI as part of non-pharmacological management and as a predictor of COPD outcomes.

Although it is a lung disease, COPD often leads to systemic complications, such as muscle wasting and sarcopenia [19]. Muscle wasting can progress to sarcopenia, a multifactorial condition characterized by progressive loss of skeletal muscle mass, impaired neuromuscular junction stability, reduced anabolic signaling, increased proteolysis, mitochondrial dysfunction, and chronic inflammation [20]. Central to its pathogenesis is decreased anabolic signaling—particularly a reduction in IGF-1–induced activation of the PI3K–Akt–mTOR axis—which leads to diminished protein synthesis, while concurrently allows FOXO transcription factors to up-regulate E3 ubiquitin ligases (Atrogin-1, MuRF-1), thus enhancing proteasomal degradation [44,45]. Mitochondrial dysfunction and oxidative stress further exacerbate atrophy, as reactive oxygen species impair mitochondrial quality control and reduce PGC-1α-mediated biogenesis. Additionally, neuromuscular junction destabilization and satellite cell senescence compromise muscle regeneration [46]. In the context of COPD, these age-associated mechanisms are significantly amplified by systemic and muscle-specific inflammation (elevated TNF-α, IL-6), chronic hypoxia, oxidative imbalances, and corticosteroid exposure—all contributing to a hypercatabolic state [47]. Crucially, COPD muscle biopsies show an overactivation of the myostatin–Smad2/3 pathway, which inhibits Akt signaling and promotes atrophy signaling via FOXO [48] (Figure 2). Concurrently, markers of autophagy and protein turnover are elevated in patients with COPD, indicating excessive degradation.

According to the European Working Group on Sarcopenia in Older People (EWGSOP), for the diagnosis of sarcopenia, three parameters should be evaluated: low muscle strength, low muscle quantity or quality, and low physical performance; moreover, if these three parameters are present together, sarcopenia can be classified as severe [21].

In general, several factors contribute to reduced muscle strength and endurance, including chronic inflammation, oxidative stress, physical inactivity, hypoxemia, hormonal imbalances, deficiencies in essential nutrients, like protein and vitamin D, and the use of systemic corticosteroids [22,23]. However, in COPD patients, sarcopenia is reported to occur in about 25% of cases and is linked to reduced lung function and a decline in overall health status [25]. Sarcopenia can negatively affect COPD patients by reducing quality of life, increasing the risk of hospitalization, and raising mortality rates [26]. The main pathogenic feature is the systemic inflammatory response with the releasing of TNFα and IL-6, which correlate negatively with grip strength (HGS) skeletal muscle mass index (SMMI) in COPD patients compared to patients without muscle loss [27]. The presence of systemic inflammation is closely linked to complications that can impact prognosis, including weight loss, cachexia, and cardiovascular disease [28]. Byun et al. examined the correlations between sarcopenia and the systemic inflammatory biomarkers IL-6 and high sensitivity (hs)TNFα. Their findings revealed significant relationships between muscle strength, as measured by HGS and muscle mass, assessed through skeletal muscle mass index (SMMI), with levels of IL-6 and hsTNFα. Multivariate analysis indicated that higher hsTNFα was a significant predictor of sarcopenia [10].

Additionally, the reduced oxygen delivery and nutrient absorption seen in advanced stages of COPD exacerbate muscle loss. The resultant sarcopenia not only diminishes physical strength and endurance but also contributes to a vicious cycle of worsening respiratory function and decreased quality of life [30]. Clinical outcomes, such as the severity of COPD, the extent of dyspnea, and the presence of CVD, are strongly related to sarcopenia in these patients [31].

Sarcopenia and HGS are linked to the frequency of AECOPD [32,33]. Perrot et al. found that the prevalence of sarcopenia was notably high among COPD patients during an acute exacerbation (48%) and remained significant after recovery (30%) [34]. Notably, De Blasio et al. reported that 58.7% of COPD patients with sarcopenia were also malnourished. This condition is common in patients with progression of COPD and, in particular, in patients with systemic inflammation, like cachectic patients. According to them, malnourished patients with sarcopenia exhibited a notable decline in BMI, fat-free mass, and HGS compared to those without sarcopenia [35,36,37,38].

Furthermore, hormonal dysregulation, including reduced levels of testosterone and insulin-like growth factor 1 (IGF-1), can hinder muscle regeneration and maintenance. These pathways illustrate the link between respiratory and muscular health in COPD, emphasizing the need for strategies targeting both lung function and muscle strength [39].

While smoking is widely acknowledged as a significant and established risk factor for COPD, the direct relationship between smoking and sarcopenia remains a subject of debate and investigation [41]. Patients with sarcopenia demonstrated lower predicted forced expiratory volume in the first second (FEV1) and had reduced exercise tolerance and lower quality of life compared to those without sarcopenia. Sarcopenia is common among individuals with COPD and has a detrimental effect on key clinical outcomes. However, further research is needed to assess its impact on mortality in this population [42].

## 4. Adipokines in Metabolic Disorders of COPD

In the last few years, it has been recognized that adipose tissue plays a central role in the control of immune and inflammatory responses with cross-talk between adipose tissue and the lung. The functional role of adipose tissue is strictly related to its endocrine functions, capable of secreting hormones known as adipokines. It is plausible that the contribution of adipose tissue in the lung inflammatory state is the molecular basis of the cross-talk between the two organs and that the secretion of adipokines is part of a very complex inflammatory milieu that is established in the lungs of COPD patients sustained by neutrophils, macrophages, and CD8^+^ T cells, with increased production of chemokines and cytokines [49].

Adiponectin and leptin are among the principal adipokines. It is believed that adiponectin is the main adipokine, being secreted in serum at very high levels between 5 and 30 μg/mL. It is generally recognized that adiponectin levels are increased in COPD patients, with even higher levels in those without bronchiectasis and worse prognosis [43]. For this reason, adiponectin has been proposed as a biomarker to identify patients with advanced COPD patients and to monitor severity and progression of the disease [43]. Functionally, such up-regulation of adiponectin levels might be traceable to multiple aspects of COPD, i.e., the sustained inflammation as well as alteration in body composition. It is proposed that the increased production of adiponectin represents a biological response in an attempt to counteract inflammation. In addition to the inflammatory response, body composition is another important factor in COPD patients, especially in controlling disease progression and prognosis. Indeed, metabolic alterations are found to be more frequent in COPD. It has been shown that BMI and fat free mass index (FFM) are inversely related to mortality in patients with COPD [50,51]. Oliveira et al. reported that adiponectin levels were significantly and positively correlated with fat mass and the fat mass index while negatively correlated with fat-free mass and the fat-free mass index in patients with bronchiectasis [52]. It has also been shown that high BMI values are associated with high levels of C-reactive protein (CRP) and a recent study identified that this association is present even in obese patients with COPD [53]. Therefore, it is plausible that systemic inflammation is one of the potential mechanisms responsible for both COPD and metabolic syndrome, as suggested by the presence of various inflammatory markers in different biological samples such as plasma, sputum, and bronchoalveolar fluid. This suggests that inflammatory response and body composition are both regulated by adipokines—with a specific regard to adiponectin—in COPD diseases participating in the control of the progression and prognosis.

Leptin is a protein product of the ob gene, synthesized and secreted mainly by white adipose tissue but expressed also by the human lung, including bronchial epithelial cells and alveolar type II pneumocytes and macrophages [54]. Systemic leptin may be associated with greater COPD prevalence and severity; Breyer et al. demonstrated a positive correlation between serum concentrations of leptin and C-reactive protein in women with COPD but not in men [55]. Schols et al. reported a positive correlation between plasma concentrations of leptin (adjusted for fat mass) and of soluble TNF receptor, a marker of systemic inflammation, among stable male patients [56]. On the other hand, other case-control studies did not confirm such an association between leptin and activity of the TNF-alpha system [57]. Such data suggest a regulation of leptin in stable/unstable COPD with a clear correlation to the control of the inflammatory response. In accordance with that, leptin concentrations rise during acute COPD exacerbations and return to baseline in the stable state following the resolution of the exacerbation [58,59]. Serum leptin levels (ng/mL) were significantly higher in obese COPD cases compared to controls and non-obese cases and during exacerbations, which indicates that leptin plays a role in the systemic inflammatory process [60]. Regarding the relationship between leptin, body weight, and composition in COPD, several studies outlined that the cause of weight loss in some patients is not due to increased circulating leptin in COPD. Instead, leptin remains regulated in COPD and further decreases in patients with low BMI, probably as a compensatory mechanism in an attempt to preserve body fat content [61]. Serum leptin hormone level is positively correlated with BMI (kg/m^2^). In this context, it is fundamental to consider the leptin/adiponectin ratio since it is recognized as a functional biomarker of adipose tissue functioning [62]. Metabolically unhealthy COPD patients have higher levels of leptin, lower levels of adiponectin, and increased insulin resistance, compared with patients without metabolic syndrome. These individuals constitute a subgroup of patients with a specific COPD phenotype characterized by an increased leptin–adiponectin imbalance and insulin resistance. Patients with stable COPD have also been shown to have increased leptin levels [63], while an increased leptin/adiponectin ratio has been reported during COPD exacerbations [59]. Besides the metabolic regulation of adiponectin in this group of COPD, Watz et al. reported that in COPD patients, the presence of metabolic syndrome is associated with an increased inflammatory profile; this coexistence would suggest a role for adiponectin in the regulation of both inflammation and body composition [64].

Adiponectin and leptin are two central hormones in COPD and can be considered as potential biomarkers for both disease severity and prognosis, as well as for the presence of concomitant metabolic disorders. The functional role of adiponectin up-regulation in the above-mentioned conditions is still a matter of debate but it is likely that this adipokine plays a role in t counteracting chronic inflammation, a hallmark of both COPD and metabolic disorders [65,66]. On the other hand, the adiponectin–leptin ratio represents a promising diagnostic and prognostic index in COPD in relation to body composition.

## 5. Targets for Lipid Related Risk: Risk Assessment Tools

Airway inflammation is a consistent feature of COPD and is implicated in the pathogenesis and progression of disease [67]. COPD is most associated with the activation of innate immune pathways, including T1 and T17 responses, but in a subset of patients—especially those with severe disease-, eosinophil-mediated, and autoimmune responses have also been implicated [5,67]. Adiponectin, an insulin sensitizing hormone, also takes part in the regulation of inflammation [68]. The serum haptoglobin level is simultaneously and significantly increased in COPD rather than in controls [69]. In addition, its expression is significantly and negatively correlated with FEV1 both in COPD and in controls, and for haptoglobin, also strongly supports the role of a pro-inflammatory cytokine in the immune systems of COPD patients [70,71]. A high proportion of patients with COPD have CVD, but there is also evidence that COPD is a risk factor for adverse outcomes in CVD [72]. This connection between COPD and cardiovascular risk appears to be linked to the increase in lipoproteins [73]. There are six major lipoproteins in the blood: chylomicrons, very low-density lipoprotein (VLDL), intermediate density lipoprotein (IDL), low-density lipoprotein (LDL); lipoprotein(a) (Lp(a)), and high-density lipoprotein (HDL) [74]. Lipoproteins in the plasma transport lipids to tissues and consist of esterified and non-esterified cholesterol, TG, and phospholipids, and protein components called apolipoproteins that act as structural components, ligands for binding to cellular receptors, and enzyme activators or inhibitors [74,75]. All ApoB-containing lipoproteins with diameter <70 nm, including the smaller TG-rich lipoproteins and their residual particles, can cross the endothelial barrier, especially in the presence of endothelial dysfunction [76] where they can remain retained in the arterial wall causing a complex process leading to lipid deposition [76,77], the initiation of an atheroma and the subsequent growth and more rapid progression of atherosclerotic plaques. Patients with severe and very severe COPD have higher levels of cholesterol and LDL in the blood and lower levels of HDL in the blood [78]. Accumulation of cholesterol and its oxidized derivatives (oxysterols) in macrophages activates the NLRP3 inflammasome, promoting the release of IL-1β and IL-18 and fostering a chronic pro-inflammatory environment [79]. Additionally, elevated membrane cholesterol enhances the formation of lipid rafts that facilitate Toll-like receptor (TLR) signaling, particularly TLR4, thereby amplifying innate immune responses [80,81]. Among lipoproteins, oxidized LDL (oxLDL) has strong pro-inflammatory effects: it induces endothelial activation, increases cytokine production, and engages TLRs, contributing to atherogenesis and systemic inflammation [82,83]. Conversely, HDL is generally anti-inflammatory, inhibiting LDL oxidation, promoting cholesterol efflux via ABCA1 and ABCG1 transporters, and suppressing inflammatory signaling; however, during aging or chronic disease, HDL becomes dysfunctional and may lose these protective effects [84,85]. ApoA-I, the main apolipoprotein in HDL, directly inhibits NF-κB signaling in macrophages [86], while enzymes, like Lp-PLA2, can further modulate the inflammatory response depending on lipid substrate availability [87]. These inflammatory and lipid-related processes are closely linked to sarcopenia. Disrupted lipid metabolism, particularly ectopic deposition of toxic lipid species, such as ceramides and diacylglycerols in skeletal muscle, impairs mitochondrial function and insulin signaling, leading to anabolic resistance and muscle protein degradation [88,89]. Systemic low-grade inflammation—often exacerbated by dyslipidemia—promotes catabolic pathways in muscle through increased cytokines like TNF-α and IL-6 [90,91]. Another potential target is represented by polyunsaturated fatty acids (PUFAs) [92]. Their biosynthetic products, including eicosanoids and docosanoids, modulate processes involved in CVD, such as inflammatory processes, endothelial dysfunction, and immune modulation [93]. Additionally, mitochondrial dysfunction and reactive oxygen species (ROS) production impair muscle homeostasis, partly by activating redox-sensitive transcription factors like NF-κB [94,95]. Insulin resistance, driven in part by lipid accumulation, further impairs muscle protein synthesis via the suppression of PI3K-Akt-mTOR signaling [96]. In a large meta-analysis of randomized trials, triglyceride lowering was associated with lower ASCVD risk, which was somewhat lower than seen for LDL-C [78,97]. In a comprehensive systematic review and meta-analysis of risk factors for premature myocardial infarction, a mild elevation of triglycerides (>150 mg/dL) was associated with a two- to three-fold increased risk of premature myocardial infarction, like that magnitude of risk noted for total cholesterol > 200 mg/dL or HDL-C < 60 mg/dL [98,99]. Furthermore, triglycerides are an important risk marker for premature CHD events in women, with a stronger risk magnitude than LDL cholesterol or non-HDL cholesterol [99,100].

## 6. Prevalence of Dyslipidemia and Cardiovascular Risk Among COPD Patients

The occurrence of CVD among COPD patients varies widely, ranging from 20% to 70%, depending on the specific type of CVD assessed [101,102,103]. A systematic review by Chen et al. found that individuals with COPD have nearly double the risk of developing CVD compared to the general population (OR 2.46, 95% CI 2.02–3.00) [11,101,104]. This elevated risk persists even after adjusting for factors such as age and smoking [101,104]. The GOLD guideline recommends the use of the National Heart and Lung Institute’s global risk calculator for evaluating cardiovascular risk in COPD patients [105]. Dyslipidemia, a common risk factor for cardiovascular disease, is also frequently seen in COPD patients, although its influence on COPD progression is not yet fully understood [103]. A large-scale study in Taiwan, which included over 100,000 patients, found that individuals with hyperlipidemia had a higher likelihood of developing COPD later on compared to those without hyperlipidemia. This finding suggests a potential link between COPD progression and endothelial damage associated with dyslipidemia [11,106]. The efficacy of statins and anti-inflammatory treatments for dyslipidemia in enhancing COPD outcomes remains a topic of debate, with observational studies showing mixed results, likely influenced by various study biases [12]. Two randomized controlled trials have specifically investigated the impact of statins on COPD: one reported no effect of simvastatin on COPD exacerbations, while the other found that simvastatin delayed the onset of first exacerbations and reduced exacerbation frequency [107]. These findings highlight the need for further research to clarify the precise role of statins in COPD management.

## 7. The Use of Medications for Dyslipidemia: Clinical Significance and Implications

According to the guidelines, statins are the main medication recommended for hypocholesterolemia [74,75]. The mechanisms of action of these drugs are to reduce cholesterol synthesis, increase hepatic expression of the LDL receptor and uptake of Apo-B-containing lipoproteins by inhibiting 3-hydroxy-3-methyl-glutaryl-CoA reductase (HMG-CoA) determining a reduction in plasma LDL-C levels and triglycerides [108,109]. Sanja et al. studied the possible effect of statin therapy on the expression of inflammatory cytokines involved in COPD; for example, IL-1β, IL-2, IL-4, IL-8, IL-10, IL-12p70, and TNF-α and the possible associations between cytokines and BMI [67]. The authors observed a reduction in these cytokines in COPD patients in treatment with statin compared to COPD patients not receiving statin therapy. In particular, COPD patients with increased BMI (>25) had reduced concentrations of IL-2 (*p* = 0.038), IL-8 (*p* = 0.039), and IL-10 (*p* = 0.005) compared to patients with normal BMI (20–25). Moreover, statin therapy is associated with the reduced expression of selected Th1 and Th2 cytokines in COPD, and this effect may be relevant in COPD patients with increased CVD [73]. A systematic review and meta-analysis by M. Abbasifard showed that statins significantly reduce serum TNF-α levels in CVD patients (SMD = −0.99 pg/mL; 95% CI: −1.43 to −0.55; *p* < 0.001) [110]. Since TNF-α and proinflammatory interleukins contribute to COPD progression [111], lowering TNF-α and IL-1, IL-8, and IL-10 may improve both systemic inflammation in COPD and cardiovascular risk. Although they are drugs with a very good safety profile, the most important side effects include myopathy, myalgia, muscle weakness, and rhabdomyolysis [75,112,113], which are dose-dependent and usually resolve with dose reduction or discontinuation of treatment [114]. A recent meta-analysis of 176 studies with 4,143,517 patients showed that the worldwide prevalence of statin intolerance, defined according to the International Lipid Expert Panel (ILEP), National Lipid Association (NLA), and Luso-Latin American Consortium (LLAC) criteria, is 9.1% [115]. In a 3-year study monitoring 774 older adults, Scott et al. [116] associated a greater decrease in muscle strength and an increased risk of falls in patients treated with statins compared to those not treated [108,116]. Therefore, for the secondary prevention of cardiovascular disease, evidence suggests that statin treatments are highly cost-effective [112,117,118,119] while their use in primary prevention is controversial [72]. Indeed, in primary prevention, 25–50% of patients with a new statin prescription discontinue therapy during the first year, with a worsening trend over time showing an adherence rate of only 25% after two years [120]. Therapeutic adherence and persistence are key factors for the success of all drug therapies.

## 8. Therapeutic Alternatives to Statins

While statins have long been considered the standard treatment for managing cholesterol levels and lowering the associated risks, many patients either do not reach their cholesterol targets or cannot tolerate statins due to side effects [121]. The muscle toxicity potentially associated with statins has been linked to a reduction in Coenzyme Q10, a crucial component of the mitochondrial respiratory chain [122]. In a three-year study involving 774 older patients, Scott et al. found that patients treated with statins experienced a greater decline in muscle strength and a higher risk of falls compared to patients who were not treated [108,115]. Recent studies have shown that non-statin cholesterol-lowering medications, such as ezetimibe and proprotein convertase subtilisin/kexin type 9 (PCSK9) inhibitors, offer cardiovascular benefits [123].

In particular, PCSK9 inhibitors are monoclonal antibodies that target PCSK9 [123], a protein involved in LDL cholesterol regulation. By binding to LDL receptors on hepatocytes, PCSK9 promotes their degradation via intracellular pathways [124]. Thus, PCSK9 inhibitors block this interaction, enhancing receptor recycling and increasing the clearance of circulating LDLcholesterol [123,124].

In this context, great interest in recent years has been around nutraceuticals and functional foods in terms of a reduction in cholesterol levels [125,126,127]. The International Lipid Expert Panel emphasized the potential use of specific nutraceuticals, such as Omega-3 Fatty Acids, plant sterols, red yeast rice, and soluble fiber, to complement standard pharmacological approaches in managing dyslipidemia. The panel highlights that while these natural compounds may help in reducing low-density lipoprotein cholesterol (LDL-C) and triglyceride levels, their efficacy can vary based on individual patient profiles, dietary habits, and genetic predispositions [128]. The potential for combining different nutraceuticals stems from the possibility of enhancing lipid-lowering effects through their complementary actions and the ability to lower the doses needed for effectiveness while ensuring tolerability [129,130]. From reviewing the literature, the combination of fiber and phytosterols in individuals with normal lipid levels and those with moderate–high cholesterol levels indicates that this combination can lead to an average reduction of about 8% in total cholesterol and 11% in LDL cholesterol. Furthermore, two studies have suggested that using both components together results in a slightly more pronounced reduction in cholesterol levels compared to using each component individually [109,131]. In this setting, Berberis aristata/Silybum marianum (BBR) is a benzylisoquinoline quaternary alkaloid which, inhibiting the proprotein convertase subtilisin/kexin type 9 (PCSK9), leads to increased levels of hepatic LDL receptors (LDLR) and reduced LDLR degradation [132,133]. Moreover, BBR activates AMP-activated protein kinase (AMPK), promoting fatty acid oxidation and inhibiting lipogenic gene expression [134,135]. A recent meta-analysis confirmed the efficacy of BBR on the reduction in the lipid level and appeared to be additive to those of statins with an improvement on glucose metabolism and blood pressure [136]. A Palimerica study evaluated the efficacy of a nutraceutical compound useful in regulating the metabolism of LDL-C and triglycerides [137]. Thanks to the combined effect of berberine, selective optichol, artichoke, fistosterols, and fenugreek, Derosa et al. demonstrated a significant control by the product not only on total cholesterol and LDL levels, but also on triglycerides and carbohydrate profile. The study included two treatment arms, one and two capsules per day, respectively. The primary objective was achieved in the treatment group with two capsules/day, in which after 3 months of treatment there was a 25% reduction in LDL-C equal to 40 mg/dl [75,137]. The guidelines state that a reduction in LDL cholesterol of 40 mg/dL reduces CV risk by about 20% [75]. Furthermore, the results of the secondary endpoints after 3 months of treatment showed a reduction of total cholesterol by 11% (27 mg/dl) in the group treated with one capsule/day and by 19% (45 mg/dl) in the group with two capsules/day, a reduction of triglycerides by 11% (19 mg/dl) vs. 16% (29 mg/dl), a reduction of glycemia by 8% (8 mg/dl) vs. 14% (13 mg/dl) and also a reduction of insulinemia by 15% (2 μUI/mL) vs. 21% (3 μUI/mL) [137].

An overview of a different therapeutic class for dyslipidemia and the role in the treatment in COPD patients is represented in Table 1 (below).

Therefore, the decision to suggest nutraceutical products to control LDL-C levels in patients in primary prevention can be taken by the physician in specific conditions [126]. In secondary prevention, the use of nutraceuticals or functional foods can be a valuable tool to reduce LDL-C levels (and, therefore, the cumulative LDL-C load) when target LDL-C levels are not reached with statins as they have an additive action. Therapeutic alternatives to statins for managing cholesterol levels and cardiovascular risk include several classes of medications and lifestyle interventions. Engaging in physical activity, especially aerobic exercise, has been shown to enhance cardiovascular outcomes by positively affecting various risk factors for CV, including dyslipidemias [109,138].

## 9. Conclusions

The complex relationship between COPD and its comorbidities underscores the importance of comprehensive management strategies that focus not only on respiratory function but also systemic complications such as muscle wasting and dyslipidemia. Statins have shown promise in reducing inflammation and improving cardiovascular outcomes in COPD patients, yet concerns regarding their side effects, especially muscle toxicity, highlight the need for alternative approaches. Therapies aimed to normalize cholesterol levels without muscle toxicity may be beneficial to avoid muscle waste progression towards sarcopenia in COPD. A comprehensive assessment and targeted optimization of nutritional status and lipid profiles are crucial to achieving effective management and improving outcomes in COPD patients. Emphasizing lifestyle modifications, integrating nutraceuticals, and exploring innovative lipid-lowering therapies could provide favorable adjuncts to traditional pharmacological treatments. However, potential confounding factors in the reviewed studies, such as rehabilitation interventions and corticosteroid use, should be considered. Furthermore, recognizing and treating dyslipidemia and sarcopenia in these patients should be a priority for improving their overall quality of life and reducing morbidity and mortality. Future studies are required to clarify the interactions between these factors to enable the development of integrated therapeutic strategies tailored to the patients with COPD.

## Figures and Tables

**Figure 1 biomedicines-13-01817-f001:**
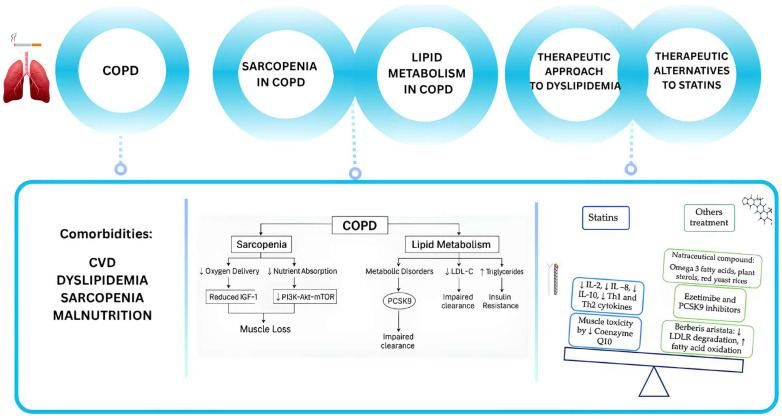
Schematic overview of the study. COPD is associated with significant comorbidities, such as cardiovascular disease (CVD), metabolic disorders, muscle wasting and sarcopenia. In particular, sarcopenia, poor nutrition, and pulmonary cachexia are associated with COPD and can lead to a worse prognosis. Sarcopenia in COPD involves reduced anabolic signaling, such as impaired IGF-1/PI3K/Akt/mTOR pathways, leading to decreased protein synthesis. Moreover, mitochondrial dysfunction and increased oxidative stress further exacerbate muscle loss. According to the guidelines, statins are the main medication recommended for hypocholesterolemia. Statin therapy is associated with a reduction in IL-2, IL-8, IL-10, Th1, and Th2 cytokines in COPD patients but the most frequent side effect (the muscle toxicity) is due to a reduction in coenzyme Q10. Alternative medications can be the nutraceuticals (like the Barberis aristata, an alkaloid that reduce LDL receptor degradation).

**Figure 2 biomedicines-13-01817-f002:**
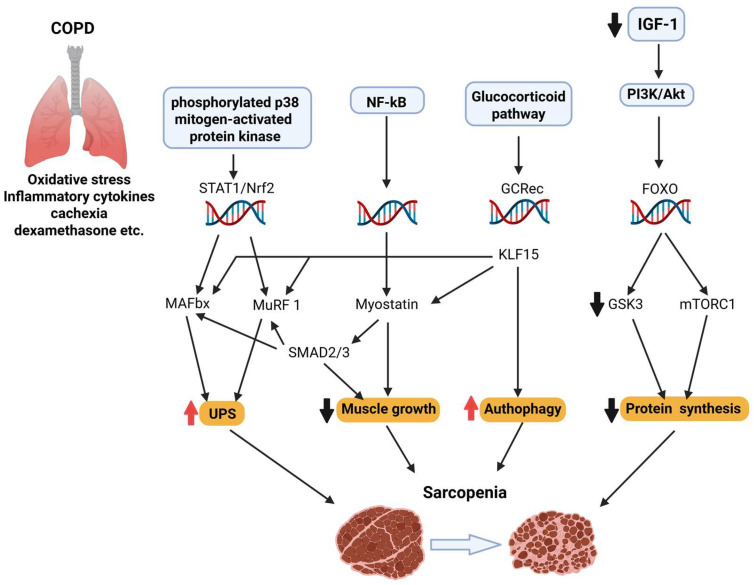
Sarcopenia is a multifactorial condition marked by the progressive loss of skeletal muscle mass and function. It involves impaired neuromuscular junction stability, reduced IGF-1/PI3K/Akt/mTOR signaling, increased FOXO-driven proteolysis via Atrogin-1 and MuRF1, mitochondrial dysfunction, and chronic inflammation. In COPD, these mechanisms are exacerbated by systemic inflammation, hypoxia, corticosteroids, and myostatin–Smad2/3 signaling, promoting a hypercatabolic state and excessive autophagy.

**Table 1 biomedicines-13-01817-t001:** Different therapeutic class for the treatment of dyslipidemia and the additional role in COPD patients. We provide a summary of different class of drugs according to the mechanism of action, lipid effects, side effects, and the correlation with real life evidence.

Therapeutic Class	Mechanism of Action	Lipid Effects	Additional Beneficts in COPD	Side Effects	Clinical Evidence
**Statins**	Inhibit HMG-CoA reductase, ↑ LDL receptor expression	↓ LDL-C, ↓ triglycerides	↓ inflammatory cytokines, possible ↓ exacerbations	Myalgia, myopathy, ↑ fall risk in elderly	Strong, but mixed on respiratory outcomes
**Ezetimibe**	Inhibits intestinal cholesterol absorption	↓ LDL-C	No known direct effects on COPD	Well tolerated	Positive results in CV prevention
**PCSK9 inhibitors**	↑ LDL receptor recycling by blocking PCSK9	↓↓↓ LDL-C (up to 50% or more)	Potential anti-inflammatory impact	Subcutaneous injections, expensive	Strong evidence in high-risk patients
**Omega-3**	Alters lipid composition, ↓ triglyceride synthesis	↓ triglycerides, modest ↓ LDL-C	Anti-inflammatory effects, endothelial function improvement	GI disconfort	Useful as adjunct therapy
**Fiber+ Phytosterols**	Inhibit cholesterol absorption in the gut	↓ LDL-C (up to -11%)	Beneficial effects on gut microbiota and metabolism	Well tolerated	Controlled studies, complementary effect
**Barberine**	↑ AMPK, ↓ PCSK9, ↑ LDLR expression	↓ LDL-C, ↓ triglycerides, improved glycemic profile	Anti-inflammatory, ↓ insulin, possible ↓ COPD exacerbations	Possible GI side effects	Positive meta-analyses, synergistic with statins
**Combined Nutraceutcals**	Berberine + phytosterols + fiber + artichoke + fenugreek	↓ LDL-C (up to -25%), ↓ total cholesterol	↓ glucose, ↓ insulin, anti-inflammatory action	Generally well tolerated	Promising Italian studies

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
