# Peer review of "Muscle Wasting and Treatment of Dyslipidemia in COPD: Implications for Patient Management"

_biomedicines, 2025, doi:10.3390/biomedicines13081817_

Round 1
Reviewer 1 Report
Comments and Suggestions for Authors
The review is devoted to the current clinical problem - sarcopenia and lipid metabolism disorders in COPD.
Comments:
- Despite a fairly detailed description of the clinical links between COPD and sarcopenia, the pathophysiological mechanisms underlying the loss of muscle mass are not detailed. There is only a brief description, which does not allow to assess the depth of the relationship between inflammation, hypoxia, metabolic disorders, and physical activity level in COPD and sarcopenia. It is recommended to describe the mechanisms of sarcopenia in general and in COPD, and to add a new figure.
- Although the aim of the article was to describe the relationship between dyslipidaemia and sarcopenia in COPD, the description of the role of cholesterol and lipoproteins is not sufficient. It is recommended to strengthen the description of the role of cholesterol in cellular mechanisms of inflammation, describe the cross-links between lipoproteins and inflammatory mechanisms, etc. It is recommended to strengthen the description of pathophysiological mechanisms linking disorders of lipid metabolism and sarcopenia.
- Figure 1 is not very informative. It is recommended to improve the figure by adding molecular mechanisms of sarcopenia, lipid metabolism disorders in COPD
- A more detailed description of the anti-inflammatory mechanisms of statins would be helpful
- The drug therapy sections would benefit from strengthening by describing other hypolipidaemic drugs including fibrates, PCSK9 inhibitors. Although there is mention of omega-3 fatty acids, but there is no description of their metabolites that play an important role in inflammation, such as specialised pro-resolving lipid mediators
Reviewer 2 Report
Comments and Suggestions for Authors
The paper provides a thorough review of the systemic complications associated with COPD, particularly focusing on muscle wasting and dyslipidemia. It effectively highlights the multifactorial nature of COPD and its impact on patient management. The paper also explores alternative lipid-lowering therapies and nutraceuticals, which is valuable for patients who may not tolerate statins well. This approach broadens the scope of treatment options available for COPD patients. And I had some comments:
1.While the paper reviews existing literature, it lacks detailed statistical analysis or meta-analysis that could strengthen the conclusions drawn about the efficacy of treatments.
2. Some sections may overgeneralize findings from specific studies without considering variations in patient populations or study designs, which could affect the applicability of the results. Some section repeated similar ideas, and I suggested to revise them or simplified.
3.Although the paper discusses the role of adiponectin and other biomarkers, it could delve deeper into the molecular mechanisms underlying these interactions to provide a more robust scientific basis.
4.The paper suggests alternative treatments to statins but may not fully address the potential biases (such as rehabilitation intervention, steroid treatment..) in the studies reviewed, which could affect the reliability of these recommendations.
5.While the paper discusses various treatments, it could benefit from a stronger focus on patient-centric outcomes, such as quality of life improvements and long-term survival rates.
Round 2
Reviewer 1 Report
Comments and Suggestions for Authors
The authors answered my questions and corrected the text, which improved its quality. In the current version, there are small typos in the text, such as ‘10.1146/annurev-med-042716-091351’ and others. It is recommended to proofread the manuscript again. It would be useful to add information on polyunsaturated fatty acids and products of their biosynthesis
Author Response
We thank the referee for the valuable comments. We have carefully considered the suggestions, thoroughly proofread the manuscript and revised the text accordingly.
Reviewer 2 Report
Comments and Suggestions for Authors
I had no further question.
Author Response
Thank you for your support and recommendations.